

**Aging and hygroscopicity variation of black carbon particles in Beijing measured by a**
**quasi-atmospheric aerosol evolution study (QUALITY) chamber**
Jianfei Peng[1, 2]*, Min Hu[1, 3]*, Song Guo[1,2], Zhuofei Du[1], Dongjie Shang[1], Jing Zheng[1], Jun
Zheng[2], Limin Zeng[1], Min Shao[1], Yusheng Wu[1], Don Collins[2], Renyi Zhang[1,2]*
[1]State Key Joint Laboratory of Environmental Simulation and Pollution Control, College of
Environmental Sciences and Engineering, Peking University, Beijing, China 100871
[2]Department of Atmospheric Sciences, Texas A&M University, College Station, Texas, 77843,
USA
[3]Beijing Innovation Center for Engineering Sciences and Advanced Technology, Peking
University, Beijing, China
*To whom correspondence should be addressed. E-mail: pengjianfeipku@gmail.com (Jianfei
Peng); minhu@pku.edu.cn (Min Hu); renyi-zhang@geos.tamu.edu (Renyi Zhang)
**Abstract.** Measurements of aging and hygroscopicity variation of black carbon (BC)
particles in Beijing were conducted using a 1.2 m$^3$ quasi-atmospheric aerosol evolution study
(QUALITY) chamber, which consisted of a bottom flow chamber where ambient air was
pulled through continuously and an upper reaction chamber where aging of BC particles
occurred. Within the reaction chamber, transmission of the solar ultraviolet irradiation was
approximately 50% - 60%, wall loss of primary gaseous pollutants was negligible, and BC
exhibited a half-lifetime about 3-7 hours. Typically, equilibrium for the primary gases,
temperature, and relative humidity between the reaction chamber and ambient air was
established within 1 hour. Rapid growth of BC particles was observed, with an average total
growth of 77±33nm and average growth rate of 26±11 nm h$^{-1}$. Secondary organic aerosol
(SOA) accounted for more than 90% of the coating mass. The O/C ratio of SOA was 0.5,
lower than the ambient level. The hygroscopic growth factor of BC particles decreased
slightly with an initial thin coating layer because of BC reconstruction, but subsequently
increased to 1.06-1.08 upon further aging. The κ (kappa) values for BC particles and coating
materials were calculated as 0.035 and 0.040 at the subsaturation and supersaturation
conditions, respectively, indicating low hygroscopicity of coated SOA on BC particles. Hence,
our results indicate that initial photochemical aging of BC particles does not appreciably alter
the particle hygroscopicity in Beijing.





## 1  Introduction

Atmospheric aerosols undergo continuous and complicated transformation during their residence time in the atmosphere. The aging of aerosols is likely resulted from both physical (i.e., coagulation, condensation, equilibrium partitioning, and evaporation) and chemical (i.e., photochemical gas-phase oxidation and multi-phase reactions) processes (Zhao et al., 2006; Qiu et al., 2013; Zhang et al., 2015). Also, there are typically large variations in the particle properties (i.e., size, mass, chemical composition, morphology, and optical and hygroscopic parameters) during aging, significantly influencing the aerosol impacts on visibility, human health, weather, and climate (Jacobson, 2001; Guo et al., 2014). A better understanding of the aging process of aerosols in the atmosphere is critical in atmospheric and climate research.

For example, the scientific interest in the climate effects of black carbon (BC) has remained, since BC is the strongest absorber of visible solar radiation (Wang et al., 2013). BC solar absorption represents a central issue in climate change research, since the synthesis of satellite, in situ, and ground observations shows that the global solar absorption (direct radiative forcing or DRF) by BC is as much as 0.9 $W \cdot m^{-2}$, second only to that of $CO_2$ (Jacobson, 2001; Bond et al., 2013; IPCC, 2013). BC also represents an important component of air pollution for large parts of the world (Zhang et al., 2015). The properties of BC are considerably modified during aging, including the size, mass, morphology, and optical and hygroscopic parameters (Khalizov et al., 2009; Xue et al., 2009a). Enhanced light absorption of BC particles during aging not only contributes to atmospheric stabilization and exacerbation of haze formation, but also imposes large positive radiative forcing on climate (Peng et al., 2016). Furthermore, the variation in hygroscopicity during aging also regulates the lifetimes of BC particles. Hygroscopic particles serve efficiently as cloud condensed nuclei (CCN), affecting the formation, longevity and albedo of clouds (Yuan et al., 2008; Wang et al., 2011). Also, deposition of BC particles, via in-cloud scavenging and wet deposition, depends highly on the particle hygroscopicity (Bond et al., 2013). In addition, the hygroscopicity also affects the aqueous-phase reactions of atmospheric pollutants (Ervens et al., 2011; Wang et al., 2016). Previous studies using hygroscopic tandem differential mobility analyzer (H-TDMA) instruments have shown that coating of hydrophilic materials significantly increases the hygroscopic growth factor of BC particles (Saathoff et al., 2003; Khalizov et al., 2009; Guo et al., 2016). The ability of BC particles to form CCN also enhanced after coating of hydrophilic materials (Kuwata et al., 2007; Tritscher et al., 2011; Ma et al., 2013; Wittbom et al., 2014). The activation supersaturation depends on the particle size, hygroscopicity of coating materials, and the coating thickness (Ma et al., 2013). The coating materials in the previous experiments include sulfuric acid (Zhang and Zhang, 2005; Khalizov et al., 2009), oxidation products from biogenic and anthropogenic hydrocarbon species (Saathoff et al., 2003; Ma et al., 2013; Khalizov et al., 2013), or SOA from single emission source (Tritscher et al., 2011). However, there still exist uncertainties for parametrization of the BC lifetime in atmospheric models, because of insufficient constraints on the hydrophobic to hydrophilic conversion of BC particles under variable ambient conditions.

Atmospheric field measurements have been performed to evaluate aging of particles on different platforms, e.g., ground, aircraft, and cruise (Moffet and Prather, 2009; DeCarlo et al.,



2010; Peng et al., 2014; Liu et al., 2015) and over different spatial scales (intensive campaigns or long-term measurements). Typically, a wide variety of state-of art instruments are employed to characterize the changes of the chemical and physical properties of aerosols. On the other hand, field measurements at fixed sites are affected by transport, local emissions, and chemistry, and quantification of the particle parameters during aging involves complex decoupling of the various processes (Peng et al., 2016). In particular, it is challenging to isolate the chemical processes from those related to meteorology (i.e., transport and mixing) and emissions.

The methods of environmental chambers or reactors have been widely employed in atmospheric chemistry research, including photochemical oxidation of volatile organic compounds (VOCs) (Zhang et al., 2000), formation and growth of aerosols (Claeys, 2004; Kalberer, 2004), nucleation of nanoparticles (Zhang et al., 2004; Wang et al., 2010; Zhang et al., 2012), aging of BC particles (Zhang et al., 2008), and cloud formation (Ruehl et al., 2016). Dependent of the scientific objectives, the designs of environmental chambers and reactors vary considerably (Zhang et al., 2015). However, few of the previous experimental methods are able to characterize the evolution of aerosols under the ambient conditions.

In this study, we present measurements of aging and hygroscopicity variation of BC particles in Beijing using a quasi-atmospheric aerosol evolution study (QUALITY) chamber (Reed, 2010; Peng et al., 2016). The performance of the QUALITY chamber for mimicking the ambient gaseous concentrations (i.e., the wall loss, and gas mixing rate), ultraviolet transmission, and meteorology parameters (i.e., temperature and relative humidity, RH) has been evaluated.

## 2   Experimental method

The 1.2 m³ QUALITY chamber was employed to study BC aging under ambient conditions (Fig. 1). The two-layer chamber was comprised of an inner layer of 0.13 mm polytetrafluoro ethylene (PFA) Teflon and an outer rigid 5.6 mm thick acrylic shell (Cyro Industries Acrylite, OP-4). Both Acrylite OP-4 Acrylic and PFA Teflon allowed for efficient ultraviolet (UV) transmission in UV-B (280-315 nm) and UV-A (315-400 nm) ranges. When exposed to sunlight, the UV light transmitted through the chamber wall and initiated photochemical reactions inside the chamber.

The two individual subdivisions of the QUALITY chamber included a bottom flow chamber, where ambient air was pulled through continuously over each experiment, and an upper reaction chamber, where aging of BC particles occurred (Fig. 1). The two chambers were separated by a 5 μm thick semi-permeable expanded polytetrafluoroethylene (e-PTFE) membrane with high degrees of chemical resistivity, microporosity, nonpolarity, and thermal stability. Gaseous species penetrated the membrane by either bulk gas flow or diffusion. The permeability of the e-PTFE was greater than 90 % for nearly all the tested volatile organic components (Fig. S1) and other gaseous pollutants, i.e., $SO_2$, $NO_x$, $O_3$ and CO. Particles, on the other hand, were blocked from penetration into the reaction chamber. The filtration efficiency of the e-PTFE membrane was measured to be over 99.5% for particles larger than 15 nm. During each chamber experiment, ambient air was pulled through the flow chamber continuously and gases in lower chamber permeated through the membrane into the reaction




chamber. Hence, an environment that continuously captured the ambient gas concentrations
without the presence of ambient particles was created inside the reaction chamber. Since the
chamber was continuously exposed to ambient gas concentrations during experiments, gases
lost due to reaction, deposition or adsorption to the seed aerosols within the reaction chamber
were steadily replenished by the exchange with the flow chamber. Several sampling ports
were set at the side of the reaction chamber for injection of seed particles or sampling during
experiment.
Seed particles were introduced in the reaction chamber via an injection line (Fig. S2). To
investigate the growth of BC particles under ambient condition, monodisperse BC particles
were injected into the QUALITY chamber. BC particles were generated by incomplete
combustion of propane fuel in a custom-made laminar diffusion burner (Santoro et al., 1983;
Qiu et al., 2012). The aerosol stream sequentially passed through a 300°C heater to evaporate
the semi-volatile organic compounds in the particle phase, a Nafion dryer to remove excess
moisture in the flow, and four one-meter-long cylinder containing both alumina spherules
coated with potassium hypermanganate and activated carbon to remove all the gaseous
pollutants (i.e., VOCs, $H_2S$, $SO_2$, $NO_x$, $O_3$). The measured removal efficiency in the cylinders
for $SO_2$, NO and $NO_2$ were 99.2%, 100%, and 99.9%, respectively. The aerosol stream was
then introduced through an ionizer and into a differential mobility analyzer (DMA, model
3081, TSI, Inc.) with stable voltage to create a monodisperse BC particles flow.
A suite of high time resolution state-of-the-art aerosol instruments simultaneously
measured a comprehensive set of BC properties throughout the BC aging process (Table S1).
The particle diameter, mass, chemical composition, hygroscopicity and ability as cloud
condensation nuclei (CCN) were measurement by a scanning mobility particle sizer (SMPS),
a differential mobility analyzer–aerosol particle mass analyzer (DMA-APM) system, a
high-resolution time-of-flight aerosol mass spectrometer (HR-ToF-AMS), a humidified
tandem differential mobility analyzer (HTDMA) system, and a could condensation nuclei
counter (CCNC), respectively. Detail information of the instruments is provided in the
supplementary material and previous publications (DeCarlo et al., 2006; Khalizov et al.,
2009). Specific measurement procedures in this study are discussed below.
**DMA-APM measurement.** The DMA-APM was used to measure the effective density of
BC particles (Pagels et al., 2009). Before any DMA-APM measurement, a SMPS scan was
made to obtain the size distribution of particles inside the chamber. The particle size
distribution was then fitted with a lognormal Gaussian distribution to derive the peak
diameter. During a DMA-APM measurement, the aerosol flow passed through DMA with a
fixed voltage to select particles with a fixed diameter. The APM then measured the mass
distribution of the selected particles with the same diameter, and the effective density of these
particles was obtained by fitting the mass distribution with a normal Gaussian distribution.
**DMA-CCN measurement.** Similar to the density measurements, CCN activation was also
measured on the basis of monodisperse particles with a peak diameter. A DMA with a fixed
voltage selected particles with a peak diameter. Both a CPC and a CCN counter were placed
in parallel after the DMA to simultaneously measure the total particle number concentration
($N_{cn}$) as well as the activated number concentration ($N_{ccn}$) at a fixed supersaturation. The





activation fraction of the BC particles with peak diameter is calculated as:
$$f_{CCN} = \frac{N_{CN}}{N_{CCN}} \qquad (1)$$

Several gradients of supersaturation were set for the CCN counter, with each one being 6
min. This method yielded a steeper curve of the CCN activation rate, which was employed to
estimate the particle diameter with 50% activation fraction ($D_{50}$) and the kappa value.
Prior to each experiment, the QUALITY chamber was flushed by zero air for more than
40 hours to ensure a clean condition and covered with two layers of anti-UV cloth to shield it
from sunlight. In the beginning of each experiment, monodisperse BC particles were
introduced into the chamber. The injection of BC particles typically lasted for 1 to 2 hours.
During the injection period, zero air passed through the bottom chamber continually to
remove any possible remaining gaseous pollutant that were removed by the activated carbon.
After the injection, ambient air was pulled through the bottom chamber at a flow rate of about
50 L min$^{-1}$ for at least half an hour in order to produce a quasi-ambient condition inside the
chamber. Finally, the anti-UV cloth was removed, and BC particles underwent aging inside
the reaction chamber. A charged zero air stream continuously passed through the space
between the two chamber layers to reduce particle wall loss. Various properties of BC
particles, including the particle diameter, mass, chemical composition, hygroscopicity, and
optical coefficients, were simultaneously measured by a suite of state-of-art aerosol
instruments in every 0.5 - 1 hour. Ambient particles and chamber particles were measured
alternately every 30 min. The aging experiments lasted for about 2-6 hours depending on the
initial BC concentrations and ambient conditions.
The BC aging experiments were conducted from August 18[th] to October 17[th], 2013 at an
urban site (PKUERS) located on the campus of Peking University in the northwestern Beijing
(39.99°N, 116.31°E) (Hu et al., 2012).
**3.     Characterization and Validation of the QUALITY chamber**
**3.1     Wall loss of gases and aerosols**
To evaluate the wall loss of both particles and gases in the QUALITY chamber, different
gaseous pollutants and particles were introduced into the chamber separately and the decay of
their concentration inside the chamber was measured by gas analyzers and SMPS,
respectively. All of the ports connected to the ambient air were closed to ensure an enclosed
system in the reaction chamber.
Particles with different chemical composition exhibited different wall loss rates.
Monodispersed BC particles with different diameters showed a small half-lifetime (τ) of
about 4-7 hours (Reed, 2010). Aerosol nucleation also occurred inside the chamber likely
from organic species (Zhao et al., 2009), which corresponded to a half time of about 3.5
hours (Fig. S3), because the nucleated particles inside the chamber was neutral with a slow
electrostatic loss to the wall.
Gas species had a longer residence time inside the chamber. Toluene and isoprene did
not show obvious wall loss during a two-day experiment (Reed, 2010). O$_3$, SO$_2$ and NO$_x$





decreased by 50% inside the chamber after more than 20 hours, suggesting slow loss. Since
the loss rate of these primary gaseous pollutants was much slower than the gas exchange rate
between the chamber, the loss of gases was replenished by the exchange with the flow
chamber.

### 3.2 UV transmission

The QUALITY chamber contained two layers of walls, an acrylic shell layer and a PFA
Teflon layer. Since the QUALITY chamber used sunlight as the photochemical origin, the
transmission spectra of the two-layer walls was of great important for the photochemical
reactions inside the reaction chamber.
The transmission efficiencies of each material were measured using a Fourier transform
infrared spectroscopy (FTIR) system. As shown in Figure 2, the Teflon film exhibited stable
transmission efficiency of about 60% in the focused wavelength range. The Acrylic shell,
however, showed very low transmission efficiency when the wavelength range was shorter
than 270 nm, and high transmission efficiency (nearly 90%) when the wavelength range was
longer than 300 nm. In general, approximately 60% of the UVA irradiation (315-400 nm
range) and 50% of the UVB irradiation (280-315 nm range) penetrated through the chamber
walls, allowing photochemical processes to take place in the upper reaction chamber.
A $NO_2$-photolysis experiment was also conducted to characterize the UV transmission of
QUALITY chamber. $NO_2$ was introduced inside the chamber at a clean and sunny day.
Reactions among $NO_2$, NO and $O_3$ will occur as the following equations:

$$NO + O_3 \rightarrow NO_2 + O_2 \qquad\qquad (2)$$

$$NO_2 \overset{hv}{\rightarrow} NO + O \qquad\qquad (3)$$

$$O + O_2 \rightarrow O_3 \qquad\qquad (4)$$

By simultaneously monitoring the concentrations of $NO_2$, NO and $O_3$ inside the chamber,
the photolysis constant of $NO_2$, $J(NO_2)$, inside the chamber was estimated. The photolysis
constant of $NO_2$ inside the chamber was on average 55% of that in the ambient air, in
agreement with the transmission spectra measurement, further confirming that the two-layer
chamber walls allowed 50% - 60% of solar irradiation in the UV range into the reaction
chamber.

### 3.3 Exchange of gases between the reaction chamber and ambient air

Additional experiments were conducted to characterize the exchange time scale for gases
in the QUALITY chamber. The chamber was firstly cleaned and flushed with zero air for 40
hours. Ambient air was then pulled through the bottom chamber (Fig. 3), and the
concentrations of gaseous pollutants, including $O_3$, $NO_x$, CO, and $SO_2$, were measured
alternatively in the ambient air and in the upper chamber. In the beginning of the experiment
the concentrations of all gaseous species were lower than those in the ambient air. After
ambient air was pulled through the bottom chamber (labeled as the black dots in Fig. 3), the
concentrations of the gaseous pollutants in the reaction chamber increased sharply. For





example, the CO concentration inside the chamber was approximate 70% of that the ambient
concentration after 30-min mixing. The concentrations inside the chamber and ambient air
exhibited little difference after 1-hour mixing (Fig. 3c). Hence, the QUALITY chamber well
replicated the ambient gas concentrations. The gas exchange rate between the bottom and
reaction chambers was calculated to be approximately 0.06 min$^{-1}$.
**3.4      Temperature and RH**
The greenhouse effect for an outdoor chamber typically increases the temperature and
decreases the RH inside the chamber. For the QUALITY chamber, however, heat produced
by the greenhouse effect inside the chamber was effectively taken away as the ambient air
continuously passed through the bottom chamber and exchanged with air in the upper
chamber. As illustrated in Figure 4, there was little difference in temperature or RH inside
and outside the chamber, when the chamber experiments lasted for more than 1 hour,
suggesting that the QUALITY chamber effectively captured the ambient temperature and
RH.
**3.5      Sulfuric acid production**
Though gaseous pollutants such as $SO_2$, $O_3$, $NO_x$ and VOCs penetrated into the reaction
chamber from the bottom chamber through the semi-permeable membrane, low volatile and
sticky gases, i.e., sulfuric acid, were unlikely to penetrate through the membrane (Fortner et
al., 2004). To characterize sulfuric acid production inside the chamber, we conducted a
special experiment by pulling ambient air into the bottom chamber, while injecting $SO_2$
directly into the reaction chamber. The experiment was conducted around noon on a clean
day when the $O_3$ concentration was around 50 ppb. A custom-made atmospheric pressure-ion
drift chemical ionization mass spectrometry (AP-ID-CIMS) (Fortner et al., 2004; Zheng et al.,
2010) was used to directly measure the concentration of gaseous sulfuric acid. Fig. 5 shows a
good correlation between $SO_2$ and gaseous $H_2SO_4$ inside the chamber, suggesting that the
QUALITY chamber well simulated the formation of low volatile gaseous species and hence
the photochemical processes.
**4.      BC evolution in the QUALITY chamber**
Time series of the ambient PM, gas, and meteorology parameters of ambient air during
chamber experiment period are illustrated in Figure 6. Except for the last experiment
conducted on October 17th, all experiments were conducted between August 18th, 2013 and
September 21st, 2013. During this period in Beijing, strong photochemical reactions and
frequent heavy pollution events occurred (Huang et al., 2010; Zhao et al., 2013). Temperature
and RH during this period (August 18th to September 21st) varied from 24°C to 38°C and
from 20 % to 90 %, respectively. The average concentration of $PM_{2.5}$, $SO_2$ and $NO_x$ were 60
± 45 µg m$^{-3}$, 3.2 ± 2.6 ppb and 33.9 ± 20.9 ppb, respectively.
The red shaded areas in Figure 6 represent the period of the nine chamber experiments.
In general, chamber experiments were conducted in the afternoon of relatively clean and
sunny days, when strong solar radiation led to fast photochemical reactions. There were two
experiments conducted under polluted days, e.g., the experiments on October 22rd and
September 11th. Table 1 summaries the conditions of the experiments. Totally, 10 BC aging





experiments were conducted, including four experiments using BC particles with initial
mobility diameter of 100 nm, three experiments using 150 nm particles, and three using 220
nm BC particles. The average concentrations of $PM_{2.5}$ and $NO_x$ over each chamber
experiment were only 9 to 69 μg m$^{-3}$ and 9 to 41 ppb, respectively. The concentrations of
VOCs, such as toluene and m/p-xylene were relatively low during most of the experiments
compared with severe pollution episodes in Beijing (Guo et al., 2014). $J(O^1D)$ and $O_3$
exhibited higher values during the chamber experiments. The average $J(O^1D)$ values and $O_3$
concentrations ranged from $3.2 \times 10^{-6}$ s$^{-1}$ to $21.1 \times 10^{-6}$ s$^{-1}$ and 26 to 92 ppb, respectively.

### 4.1    BC growth

To quantify the growth of BC particles, several parameters were used to describe the
properties of BC particle, including the effective density, mobility diameter and mass
equivalent diameter. Material density ($\rho_m$) is the average density of the solid and liquid
material in the particle. Assuming that the volume of the species coexisting in an individual
particle does not change upon mixing, the density of an internally mixed particle is calculated
using the material densities and mass concentrations of particulate constituents (DeCarlo et
al., 2004; Pagels et al., 2009),
$$\rho_m = \frac{\sum_{species} m_i}{\sum_{species} V_i} = \frac{\sum_{species} m_i}{\sum_{species} \frac{m_i}{\rho_i}} = \frac{\sum_{species} MC_i}{\sum_{species} \frac{MC_i}{\rho_i}} \tag{5}$$

where $\rho_i$ is the material density of species $i$, $V_i$ is its volume, $m_i$ is its mass, and $MC_i$ is its
mass concentration. This approach is based on the assumption that there is no void space
enclosed within the particle envelope. Hence, the material density is larger than the true
particle density with internal voids in particles. In this study, $\rho_m$ is calculated from the
chemical composition of coating materials measured by AMS. A value of 1.35 for the
material density of SOA formed during chamber experiment was obtained by directly
measuring the density of newly form particles inside the chamber via DMA-APM system.
Effective density ($\rho_{eff}$) is defined as the ratio of the measured particle mass ($m_p$) to the
particle volume calculated assuming a spherical particle with a diameter equal to the
measured mobility diameter ($D_m$) (DeCarlo et al., 2004; Pagels et al., 2009; Xue et al.,
2009b):
$$\rho_{eff} = \frac{6m_p}{\pi D_m^3} \tag{6}$$

In this study, $m_p$ of BC particles was measured by the APM and mobility diameter ($D_m$)
was measured by the DMA. The effective density reflects the information on both particle
density and shape. If particles are spherical in the absence of internal void, the effective
density equals the material density. If particles are non-spherical, the calculated volume and
volume concentration are larger than the true values, and the effective density is less than true
particle and the material density.
Figure 7 exhibits the change of particle density and diameter in three typical BC aging
experiments using BC particles with the initial diameter of 100 nm, 150 nm, and 220 nm




(experiment #4, #5, and # 9 in Table 1, respectively). The average $PM_{2.5}$ concentrations in
these experiments were 40, 27, and 12 μg m$^{-3}$, respectively, suggesting relatively clean
conditions during the experiments. In all three experiments, aging of BC particles occurred
between 13:00-14:00 and 17:00 in the afternoon. The highest $J(O^1D)$ value varied from 1.7 to
2.4 × 10$^{-5}$ s$^{-1}$ and decreased generally over the experiment period. Average $O_3$ concentrations
during the three experiments were 68, 83 and 54 ppb, respectively, indicating strong
oxidation during the experiment periods.
The initial $\rho_{eff}$ values of BC particles were 0.46, 0.34 and 0.25 g cm$^{-3}$ for particles with
the initial diameter of 100 nm, 150 nm, and 220 nm, respectively, indicating fractal BC
aggregates with chain-like branches (Zhang et al., 2008). After aging of 1 hour, $\rho_{eff}$ of BC
particles in the three experiments increased to approximately 1.2 g cm$^{-3}$, suggesting that
formation of the secondary components changed the morphology from chain-like BC
particles into a more compact shape (Zhang et al., 2008; Peng et al., 2016). The morphology
change was further confirmed by a decrease of $D_m$, particularly for larger BC particles that
were more fractal than smaller BC particles.
As the $D_m$ is largely influenced by the particle morphology, we utilized the parameter of
mass equivalent diameter ($D_{me}$) to describe the growth of particles. Based on the mobility
diameter ($D_m$), material density ($\rho_m$) and effective density ($\rho_{eff}$), the $D_{me}$ is calculated
assuming that particles are compact and with a spherical morphology (DeCarlo et al., 2004):
$$D_{me} = \sqrt[3]{\frac{\rho_{eff}}{\rho_m}} D_m \qquad\qquad (7)$$
The change in the mass equivalent diameter ($\Delta D_{me}$) during BC aging is defined as the total
coating thickness, and the ratio of the total coating thickness to the initial mass equivalent
diameter ($\Delta D_{me}/D_{me,0}$) is defined as the coating fraction.
The initial $D_{me}$ of fresh BC particles with initial $D_m$ of 100, 150 and 220 nm were 61, 84
and 114 nm, respectively. In contrast to the mobility diameter, $D_{me}$ increased continuously
during the entire experiment. After 3-4 hours, $D_{me}$ in the three experiments increased to 133,
169 and 197 nm, respectively (Fig. 7), with the average growth rates of 19, 29 and 31 nm h$^{-1}$.
Higher growth rates in $D_{me}$ occurred around noontime, when the $J(O^1D)$ value was higher and
the photochemical reaction was stronger. On the other hand, much less growth rate was
observed during late afternoon or with cloud coverage (As shown in Fig. 7c at 15:00),
indicating that the growth was driven by photochemical reactions.
The increases of the particle density and diameter in all the experiments are summarized
in Table 2. Fast aging of BC particles occurred in all experiments. The total growth of $D_{me}$
ranged from 40 nm to 152 nm within 3-6 hours, with an average growth of 73 nm. The
average growth rate was 26±11 nm h$^{-1}$, demonstrating large secondary production under the
ambient conditions in Beijing. The largest growth rate ($\Delta D_{me}$ = 152 nm) was observed in
experiment #8, when solar irradiation was the strongest among all experiments (Table 2).
Correlation analysis was made between the average growth rate of BC particles ($\Delta D_{me}/\Delta t$)
with $O_3$, $PM_{2.5}$, $J(O^1D)$, and temperature during the different experiments (Fig. S4). The



growth rate of BC particles exhibits no correlation with $O_3$ concentration ($R^2$=0.00), weak
negative correlation with $PM_{2.5}$ concentration ($R^2 = 0.25$), and strong positive correlations
with $J(O^1D)$ ($R^2 = 0.80$) and temperature ($R^2 = 0.67$), indicating the importance of
photochemical production on the BC coating materials.

### 4.2     Chemical composition of coating materials

Particle composition measurements by AMS during chamber experiments reveal a
majority of coating materials (above 90%) as SOA (Fig. 7). The concentration of SOA inside
the chamber reached up to 9 $\mu g\ m^{-3}$ in several experiments, suggesting fast formation of SOA
via gas phase oxidation of VOCs. The SOA formation in Beijing is likely attributed to a large
amount of anthropogenic aromatic VOCs (Peng et al., 2017).
The elemental compositions of OA inside the chamber, i.e., the oxygen to carbon (O/C)
ratio and the hydrogen to carbon (H/C) ratio, were calculated based on the updated ambient
calibrations (He et al., 2011; Canagaratna et al., 2015). The H/C and O/C ratios of organics
for coating on BC particles exhibit notable trends during the aging process. Figure 8A shows
an example of the evolution of H/C and O/C ratios in experiment #8. The data were corrected
for the $CO_2$ concentration in the chamber, which were introduced into the chamber with BC
particles and influenced the abundance of $m/z = 28$ and 44 in the AMS mass spectra. The H/C
ratio decreased from 1.73 to 1.45 over six hours. Accordingly, the O/C ratio increased from
0.32 to 0.50 during the same time, revealing that further oxidation of SOA occurred in the
latter part of the experiment. The lower final O/C ratio in the chamber experiment (0.5) than
that under the ambient conditions (Hu et al., 2016) implies that there is oxidation on a longer
timescale or by the aqueous pathway for the formation of highly oxidized SOA in the ambient
air (Zhang et al., 2015).
Furthermore, the mass spectra of OA inside the chamber shows strong correlation with
the less-oxidized oxygenated organic aerosols (LO-OOA) derived from field measurements
in Beijing (Hu et al., 2016), which likely arose from oxidation of aromatic VOCs emitted
from vehicles (Peng et a., 2017). The correlation coefficient ($R^2$) initially was 0.88 and raised
to 0.99 sharply (Fig. 8B), indicating that the chamber well simulated the formation of
LO-OOA.
In our study, the secondary inorganic aerosols, i.e., sulfate, nitrate and ammonium, only
accounted for less than 10% of the coating materials on BC particles. This is consistent with
the previous studies showing that the concentration of organics is much larger than those of
sulfate and nitrate during the early stage of haze development in Beijing (Guo et al., 2014).
The low observed sulfate concentration in this study suggests that the gas phase formation of
sulfuric acid was unimportant under our experimental conditions. On the other hand, it has
been shown that the aqueous-phase reactions represent the dominate pathway for sulfate
formation in Beijing (Guo et al., 2010; Wang et al., 2016).
$NO_2$ has a higher reaction coefficient with the OH radical ($8 \times 10^{-12}\ cm^3\ molecule^{-1}\ s^{-1}$)
than $SO_2$ (Zhang et al., 2015). Nitrate acid formed in the gas phase is transformed into nitrate
salts by the reaction with ammonia in the equilibrium process:
$$HNO_3(g) + NH_3(g) \leftrightarrow NH_4NO_3(s) \tag{8}$$




The equilibrium of this reaction is highly depended on ambient temperature and RH (Zheng
et al., 2008). In this study, chamber experiments were conducted in the afternoon with high
temperature and low RH (Table 1), which shifted the thermodynamic equilibrium to the gas
phase.

### 4.3  Hygroscopicity evolution

***HTDMA measurement***

The hygroscopic growth factors (HGF) of particles in each experiment were
continuously measured by the HTDMA system and corrected for the reference "dry"
diameters,

$$HGF = \frac{D_{wet,t}/D_{m,t}}{D_{dry,0}/D_{m,0}} \qquad (9)$$

where $D_m$ is the mobility diameter of fresh or coated particles at dry condition, $D_{dry}$ is the
mobility diameter of particles after experiencing a low humidity (below 30%) cycle in
HTDMA, and $D_{wet}$ is the mobility diameter of particles after experiencing a high humidity
cycle (87%) in HTDMA.
Figure 9 shows the hygroscopicity variation of BC particles with the initial mobility
diameter ($D_{m,0}$) of 100 nm and 150 nm. The measured HGF of 0.999 - 1.004 for fresh BC
particles suggests high hydrophobicity, consistent with the previous studies (Khalizov et al.,
2009;Weingartner et al., 1997). After exposed to sunlight and ambient gaseous pollutants for
several hours, the HGF of these BC particles increased to 1.02-1.08 at the end of each
experiment. The HGF value varied with the total growth ($\Delta D_{me}$) of BC particles, but was
constant at the same $\Delta D_{me}$ for different experiments (Fig.9). The final HGF values shown in
Figure 9 (1.02-1.08) were much lower than those in previous laboratory studies (Khalizov et
al., 2009;Tritscher et al., 2011) but similar to the low hygroscopic fraction in field
observations (Swietlicki et al., 2008), even for growth of particle size up to 90 nm in our
experiments.
The HGF is affected by many factors, e.g., the particle chemical composition and
morphology as well as RH (Qiu et al., 2012). The hygroscopicity of BC particles coated with
inorganic components, i.e., sulfuric acid (Khalizov et al., 2009), is significantly higher than
that coated by organic compounds (Tritscher et al., 2011). In this study, the major component
of the coating substance was LO-OOA with a O/C ratio about 0.5. The low oxygen content of
SOA coated on BC particles explains the low hygroscopicity (Jimenez et al., 2009),
indicating that coating of BC particles during the early stage haze development in Beijing
does not considerably increase the particle hygroscopicity.
The morphology of BC particles directly affects the HGF. As illustrated in Figure 10,
when the $\Delta D_{me}$ was 18 nm and 22 nm for 100 nm and 150 nm BC particles, respectively, the
HGF decreased slightly to about 0.99, suggesting that a thin layer of coatings on BC particles
decreased the particle diameter, even though a certain amount of water absorbed by BC
particles increased the particle mass. The surface tension of the water layer produced an




inward force on the "chain-like" branches of BC particles, leading to particle reconstruction,
and a more compact morphology. Such change was also identified in laboratory studies
(Weingartner et al., 1997; Tritscher et al., 2011; Qiu et al., 2012). In this study, the BC
particles became spherical when $\Delta D_{me}$ was 30 nm and 40 nm for particles with initial $D_m$ of
100 nm and 150 nm, respectively (Peng et al., 2016). Therefore, when $\Delta D_{me}$ was large, the
HGF value was not influenced by reconstruction.
### *CCN measurements and κ closure*
The CCN activation faction ($f_{CCN}$) of BC particles at different supersaturation during two
typical experiments is illustrated in Figure 10. Fresh BC particles were not activated even at
very high supersaturation conditions (0.7%). With aging, $f_{CCN}$ rapidly raised to nearly 100%
at high supersaturation (0.7% for experiment #4 and 0.6% for experiment #6). After several
hours, BC particles became CCN at lower supersaturation. The $f_{CCN}$ at 0.4 supersaturation
(Fig. 11a in experiment #4) and 0.3 supersaturation (Fig. 11b in experiment #6) exceeded 50%
before the end of these two experiments, suggesting that aging increases the ability of BC
particles to become CCN (Wittbom et al., 2014) and a large amount of coatings results in
activation at lower supersaturation.
To further investigate the hygroscopicity of BC particles and combine the measurements
using HTDMA and CCN, we evaluated the hygroscopicity parameter, kappa (κ) (Petters and
Kreidenweis, 2007). The approximate relationship between the dry particle mass equivalent
diameter ($D_{me}$), the critical saturation ratio (Sc) and the apparent κ value of particles is
describe as:
$$\kappa = \frac{4A^3}{27D_{me}\ln^2 S_c} \tag{10}$$

where A is a parameter that includes several features of the solvent,
$$A = \frac{4\sigma_{s/a}M_w}{RT\rho_w} \tag{11}$$

$M_w$ is the molecular weight of water, $\rho_w$ is the density of water, $\sigma_{s/a}$ is the surface tension of
the solution/air interface, R is the universal gas constant, and T is temperature.
In addition to the supersaturated condition, the κ theory also adopts the form for the
subsaturated condition, using the HGF from HTDMA measurement and RH:
$$\frac{RH}{exp\left(\frac{A}{D_{wet}}\right)} = \frac{HGF^3 - 1}{HGF^3 - (1 - \kappa)} \tag{12}$$

where $D_{wet}$ is the wet diameter of particles.
The apparent κ values of BC particles calculated by HTDMA (κ_HTDMA) and CCN
(κ_CCN) are shown in Figure 10. The κ of fresh BC particles was near zero. With aging,
SOA coated on BC particles increased the κ_ HTDMA and κ_ CCN to approximately 0.04,
although the κ_ HTDMA and κ_ CCN exhibited difference features. A slightly higher κ_CCN
than κ_HTDMA at the beginning of aging was identified, attributed to reconstruction of BC





particles after humidified and underestimation of HGF and thus the κ_HTDMA value. Such a
difference between κ_CCN and κ_HTDMA was also observed in previous studies (Tritscher
et al., 2011; Martin et al., 2013). Nevertheless, the apparent κ values from two both methods
were comparable at the end of both experiments.

Assuming that a simple mixing rule is applicable to coated BC particles, the κ for
coating materials can be calculated based on the volume fraction of BC and SOA:
$$\kappa = \sum_i \varepsilon_i \kappa_i \qquad\qquad (13)$$
where $\varepsilon_i$ represents the volume fraction of species $i$.

The κ values of the coating materials were 0.04 at the end of our experiments for both
CCN and HTDMA method, much lower than that of ambient aerosols in Beijing (Gunthe et
al., 2011; Wu et al., 2016) and those of SOA in previous chamber studies (Jimenez et al.,
2009; Tritscher et al., 2011; Martin et al., 2013). As discussed above, the coating substances
on BC particles were mainly SOA formed from photochemical oxidation. The κ of SOA
depends on the oxidation degree, which is correlated to the O/C ratio (Jimenez et al., 2009).
The O/C ratio of the coating SOA was 0.5 in our experiment, likely explaining the low κ with
LO-OOA.
**5    Conclusions**

In this paper, we present measurements of aging and hygroscopicity of BC particles in
Beijing using the QUALITY chamber. The unique two sub-chamber design facilitates the
evaluation of aging of BC particles under ambient conditions, by mimicking the ambient
gaseous concentrations without the presence of ambient aerosols. High UV transmission
efficiency (50-60%) and negligible wall loss of primary gaseous pollutants are shown for the
chamber performance. The validation experiments demonstrate little differences in the
primary gas concentrations, temperature, and RH between the chamber and the atmosphere,
suggesting that the chamber captures the evolution of ambient conditions. In addition, our
results show sulfuric acid production correlated with $SO_2$, indicating that the chamber well
simulates photochemical-driven formation of low volatile gaseous species by the hydroxyl
radical.

BC aging experiments were performed using the QUALITY chamber in Beijing. Fast
growth of BC particles (on average 26±11 nm h$^{-1}$) was observed, and SOA was identified as
the dominate component of the coating materials on BC particles, while inorganic species,
such as sulfate and nitrate, were unimportant under our experimental condition and timescale.

The HGF of BC particles exhibited a very low value (1.02-1.08) after several hours aging.
A slight decrease of HGF with a thin coating layer indicated reconstruction of BC particles
after humidified. Also, a very low kappa value (0.035) for BC particles at both subsaturation
and supersaturation conditions were found, with HTDMA and CCN measurements. Hence,
our results indicate that initial photochemical aging of BC particles does not appreciably alter
the particle hygroscopicity in Beijing.

**Acknowledgement**





This work was supported by National Natural Science Foundation of China (91544214,
41421064), the National Basic Research Program, China Ministry of Science and Technology
(Grant 2013CB228503), National Natural Science Foundation of China (Grant 21190052),
and the China Ministry of Environmental Protection's Special Funds for Scientific Research
on Public Welfare (Grant 20130916). R.Z. acknowledged support from the Robert A. Welch
Foundation (Grant A-1417) and Houston Advanced Research Center. We thanked Wei Hu and
Zhaoheng Gong for their assistance with the AMS data analysis, Wentai Chen and Yue Li for
providing VOCs data.





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





**Table 1.** Summary of ambient conditions for BC aging experiments conducted in Beijing. The $PM_{2.5}$, gas concentrations and meteorological condition were averaged from the entire experimental time. $D_m$ and T represent the mobility diameter and temperature, respectively.

| No. | Date | Time | | Initial $D_m$ (nm) | $PM_{2.5}$ (µg m$^{-3}$) | $J(O^1D)$ ($10^{-6}$) | Gas Concentration (ppb) | | | | | | Meteorological Conditions | |
|-----|------|------|------|------|------|------|------|------|------|------|------|------|------|------|
| | | Start | End | | | | Toluene | Xylene | $SO_2$ | $NO_x$ | $O_3$ | T (°C) | RH (%) |
| #1 | Aug. 18th | 12:41 | 14:47 | 95 | 43 | 19 | 0.49 | 0.13 | 1.7 | 8.8 | 56 | 35 | 27 |
| #2 | Aug. 22nd | 13:32 | 16:51 | 96 | 69 | 3.7 | 3.41 | 0.78 | 2 | 36.1 | 26 | 26 | 69 |
| #3 | Sep. 7th | 12:40 | 14:52 | 97 | 12 | 17.5 | 0.71 | 0.17 | 2.7 | 10.2 | 75 | 30 | 35 |
| #4 | Sep. 9th | 13:13 | 17:06 | 97 | 40 | 6.3 | 0.77 | 0.21 | 4 | 11 | 68 | 26 | 50 |
| #5 | Sep. 1st | 13:19 | 16:13 | 147 | 27 | 11.6 | 0.76 | 0.19 | 4.6 | 19.9 | 83 | 31 | 33 |
| #6 | Sep. 11th | 13:50 | 17:25 | 147 | 57 | 6.1 | 1.57 | 0.39 | 6.7 | 17.1 | 92 | 29 | 42 |
| #7 | Sep. 21st | 15:31 | 17:41 | 146 | 30 | 2.1 | 0.75 | 0.29 | 2 | 10.6 | 90 | 28 | 37 |
| #8 | Aug. 24th | 11:37 | 16:06 | 216 | 8.8 | 21.1 | 0.98 | 0.3 | 1.7 | 15.6 | 57 | 36 | 25 |
| #9 | Sep. 5th | 14:06 | 16:44 | 220 | 12 | 8.3 | 0.45 | 0.2 | 2.2 | 14.6 | 54 | 29 | 35 |
| #10 | Oct. 17th | 12:54 | 17:13 | 224 | 57 | 3.2 | - | - | 13.8 | 41 | 34 | 18 | 30 |





**Table 2.** Summary of particle properties for BC aging experiments conducted in Beijing. $D_m$, $\rho_{eff}$ and $D_{me}$ represent the mobility diameter, effective density and mass equivalent diameter, respectively.

| No. | Date | $D_m$ | | $\rho_{eff}$ | | $D_{me}$ | | | |
|---|---|---|---|---|---|---|---|---|---|
| | | Initial (nm) | Final (nm) | Initial (g cm⁻³) | Final (g cm⁻³) | Initial (nm) | Final (nm) | $\Delta D_{me}$ (nm) | Growth Rate[a] (nm h⁻¹) |
| #1 | Aug. 18th | 95 | 157 | 0.50 | 1.35 | 62 | 162 | 100 | 47 |
| #2 | Aug. 22nd | 96 | 129 | 0.46 | 1.31 | 61 | 126 | 65 | 20 |
| #3 | Sep. 7th | 97 | 147 | 0.45 | 1.25 | 62 | 142 | 80 | 36 |
| #4 | Sep. 9th | 97 | 136 | 0.43 | 1.30 | 61 | 133 | 62 | 19 |
| #5 | Sep. 1st | 147 | 170 | 0.34 | 1.36 | 85 | 168 | 83 | 29 |
| #6 | Sep. 11st | 147 | 162 | 0.34 | 1.34 | 84 | 159 | 75 | 20 |
| #7 | Sep. 21st | 146 | 132 | 0.34 | 1.05 | 84 | 116 | 32 | 15 |
| #8 | Aug. 24th | 216 | 272 | 0.32 | 1.37 | 123 | 275 | 152 | 34 |
| #9 | Sep. 5th | 220 | 202 | 0.25 | 1.33 | 114 | 197 | 83 | 31 |
| #10 | Oct. 17th | 224 | 224 | 0.24 | 0.52 | 117 | 157 | 40 | 11 |
| | **Average** | | | | | | | **77±33** | **26±11** |

[a] The growth rate is calculated using the data between 12:00 and 17:00 for each experiment.





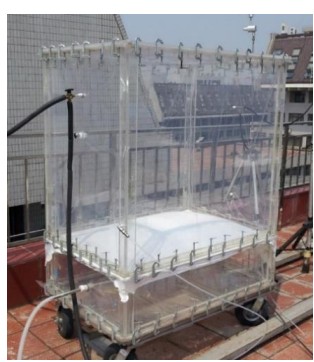 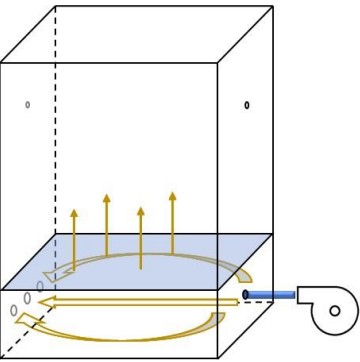

**Figure 1.** Photo (left) and schematic (right) of the quasi-atmospheric aerosol evolution study (QUALITY) chamber





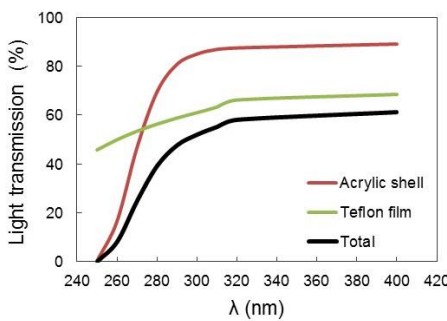

**Figure 2.** Light transmission spectra of the PFA Teflon film (yellow line), the acrylite shell (red line), and their total transmission in the UV range (black line).




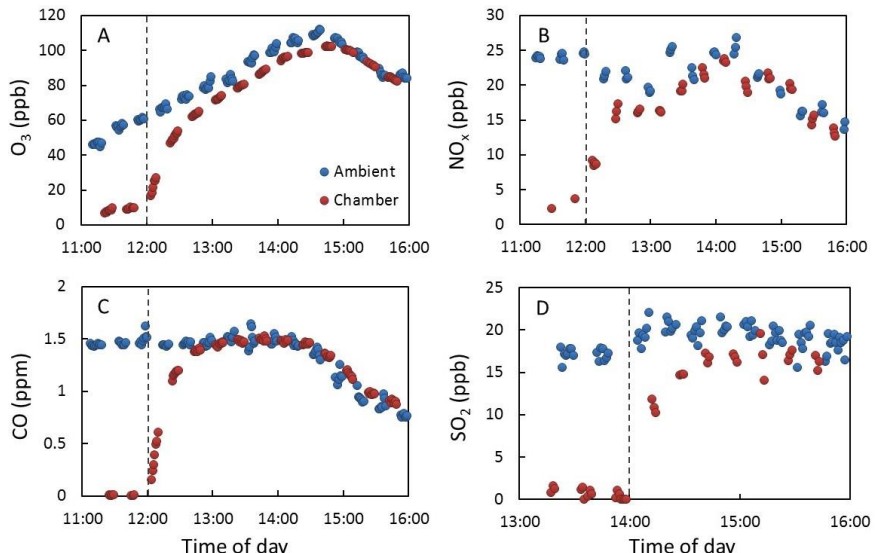

**Figure 3.** Concentrations of $O_3$ (A), $NO_x$ (B), CO (C), and $SO_2$ (D) measured inside the QUALITY chamber (Red circles) and in the ambient air (blue circles). The vertical dashed lines denote the time when the ambient air started to be pulled through the bottom flow chamber and the ambient gases began to exchange into the upper reaction chamber.





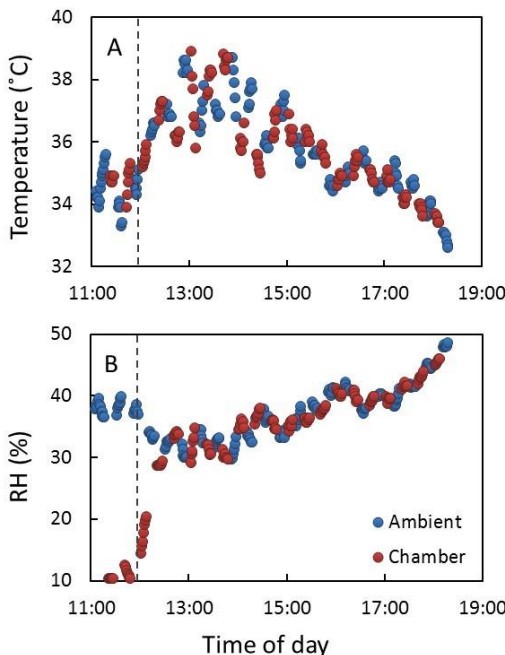

**Figure 4.** Temperature (A) and RH (B) measured inside the QUALITY chamber (Red circles) and in the ambient air (blue circles). The temperature and RH were measured by probes placed in the sampling tube adjacent to the chamber. The vertical dashed lines denote the time when the ambient air started to be pulled through the bottom flow chamber and the ambient gases began to exchange into the upper reaction chamber.



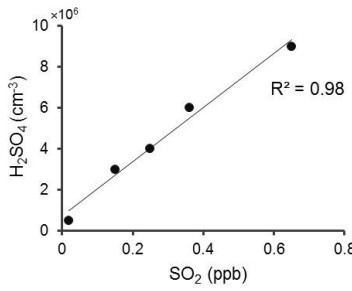

**Figure 5.** Sulfuric acid concentration as function of SO$_2$ concentration inside the chamber





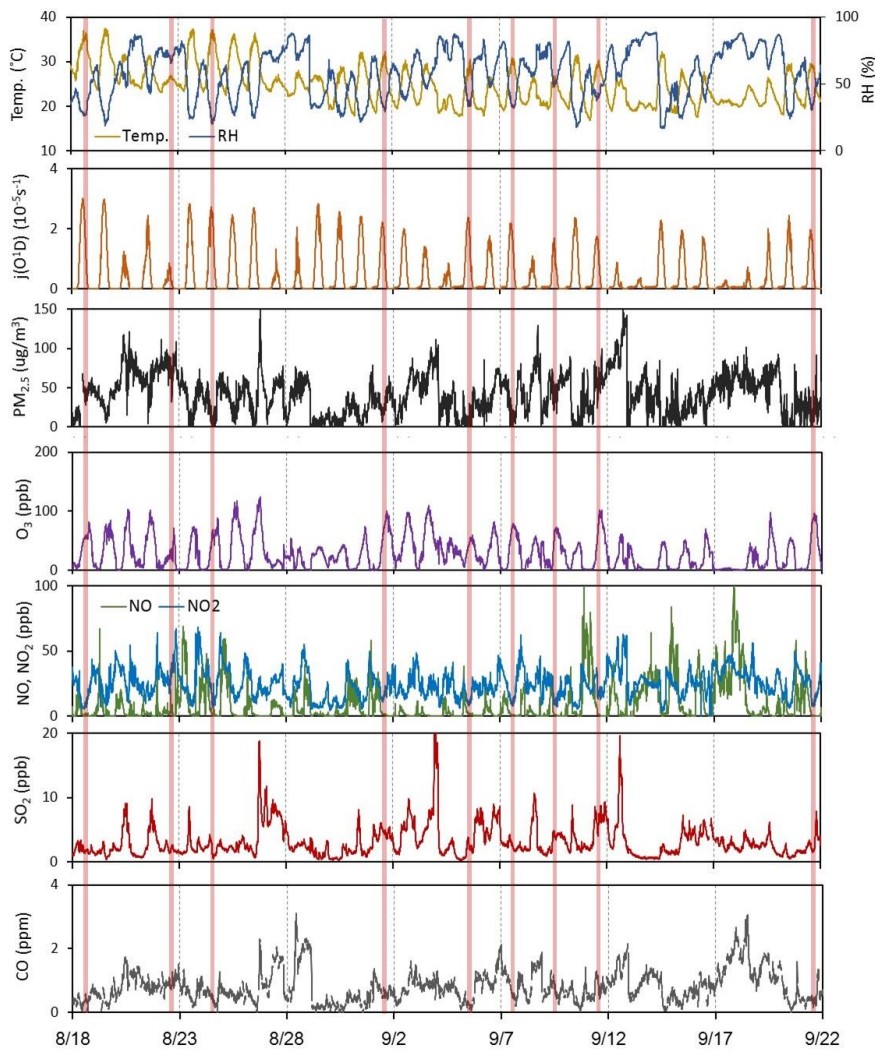

**Figure 6.** Time series of ambient pollutant concentrations and meteorological parameters during the experimental period in Beijing. Red bars indicate periods when BC aging experiments were conducted.

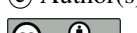



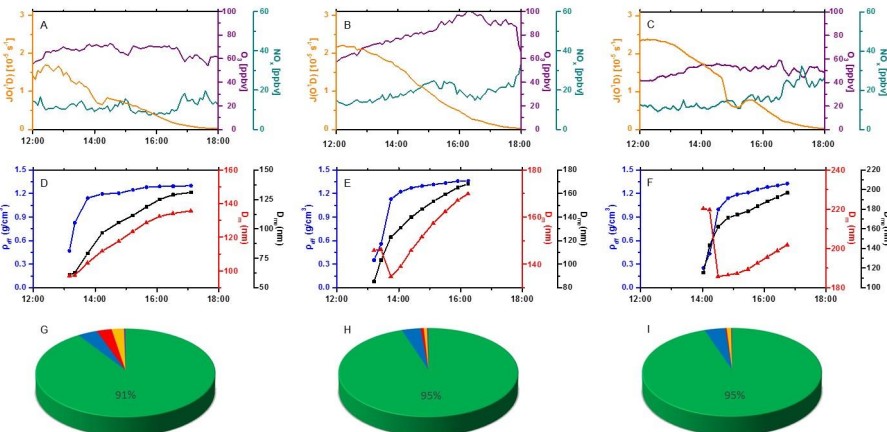

**Figure 7.** Ambient condition (A, B,C), changes of diameter and density of BC particle (D, E, F), and the chemical composition of coating materials (G, H, I) during three typical aging experiments. A and D correspond to experiment #4; B and E correspond to experiment #5, and C and E correspond to experiment #9. $D_m$ is the peak mobility diameter of BC particles, $\rho_{eff}$ is the best-fit effective density of BC particles with the mobility diameter $D_m$, $D_{me}$ is the mass equivalent diameter of BC particles, and $J(O^1D)$ represents the measured photolysis rate constant for $O(^1D)$. The colors of green, blue, red, yellow and purple in the pie charts represent organics, nitrate, sulfate, ammonium, and chlorine, respectively. The numbers in figure G, H and I are the mass fraction of organics.




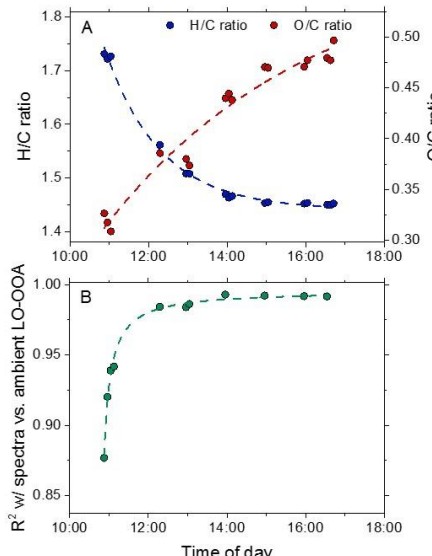

**Figure 8.** The evolution of organic aerosols inside the chamber during an aging experiment (#8) in Beijing. (A) the H/C and O/C ratios of organics on aged BC particles; (b) the correlation coefficients ($R^2$) between the evolving total OA spectra in chamber experiment and the LO-OOA spectra derived from the Beijing field data set.





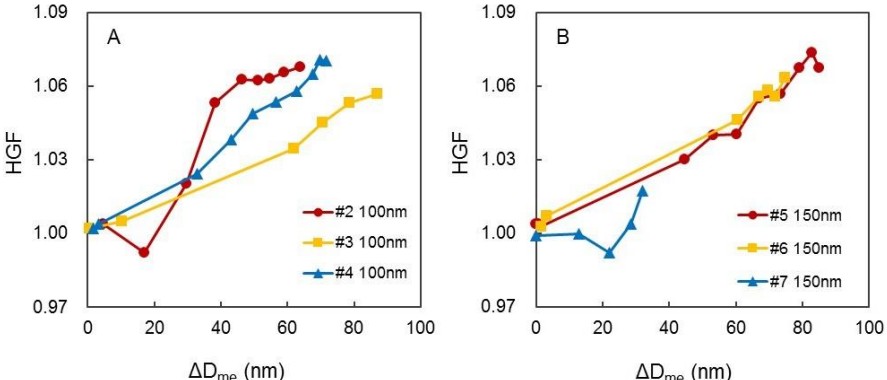

**Figure 9.** Evolution of hygroscopic growth factors (HGF) of BC particles during aging as a function of $\Delta D_{me}$. (A) three experiments with 100 nm BC particle; (B) three experiments with 150 nm BC particle. Different colors in each figure represent different experiments. Hygroscopicity measurement is not available for experiment #1.



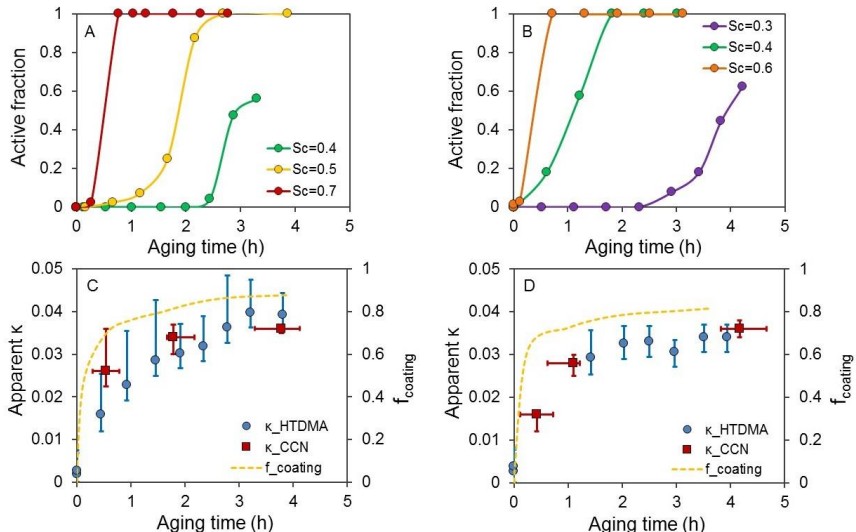

**Figure 10.** The active fraction of BC particles under diversified supersaturation (A, B) and the closure of apparent κ for BC particles with initial diameter (C, D) during aging in two typical experiments. A, C are the results from experiment #4 with 100nm BC particles and B, D represents experiment #6 with 150nm BC particles. Red squre and blue circle in C and D represent the apparent κ calculated using CCN counter data and HTDMA data, respectively. Yellow slash line represents the fraction of coating materials on BC particles. The error bars of κ_CCN and κ_HTDMA represent the uncertainty in the calculation.