# Peer review of "Aging and hygroscopicity variation of black carbon particles in Beijing measured by a quasi-atmospheric aerosol evolution study (QUALITY) chamber"

_Atmospheric Chemistry and Physics, 2017_

## Referee Comment (RC1) · Anonymous Referee #1 · 26 May 2017

In the manuscript the authors measured several physicochemical properties, including size, density, chemical composition, hygroscopicity, and Cloud Condensation Nuclei (CCN) activity of coated Black Carbon (BC) particles as a function of aging (coating with condensable species from photo-oxidation of VOCs) in a quasi-atmospheric aerosol evolution study (QUALITY) chamber using a suite of instruments, such as differential mobility analyzer, aerosol particle mass analyzer, High Resolution – Time of Flight – Aerosol Mass Spectrometer (HR-ToF-AMS), Humidified Tandem Differential Mobility Analyzer (HTDMA), and a CCN Counter (CCNC). Their results show that under ambient condition in Beijing, the BC undergoes rapid growth to 77±33 nm coating

thickness with an average growth rate of 26±11 nm h-1. The O/C ratio of the SOA coating is 0.5, lower than ambient level, indicating the lack of aqueous phase oxidation inside QUALITY chamber. The hygroscopic parameter ($\kappa$) is about 0.035 ∼ 0.040 as measured by HTDMA and CCNC, suggesting that the initial photochemical aging of BC particles does not appreciably alter the particle hygroscopicity in Beijing. This study and the data provided are quite extensive. The most valuable addition of this manuscript is the "close to ambient (Beijing)" aging condition in the QUALITY chamber. The author showed in Figure 3 that the O3, NOx, CO, and SO2 concentrations inside and outside of the chamber is close. However, the VOCs plot inside and outside of the chamber is missing. The authors also mentioned they applied heater, drier, alumina spherules coated with potassium permanganate (KMnO4), and activated charcoal to remove the VOCs, H2S, SO2, NOx, and O3. It would be nice if the authors could also show the VOCs are efficiently removed down to sub-ppb level while injecting the BC particles into the chamber. Did the authors use proton-transfer-reaction mass spectrometer (PTR-MS) to monitor the VOCs inside and outside the chamber? Since the coating is caused by condensable species from photo-oxidation of VOCs and "close to ambient" condition is an important part of the manuscript, I would suggest the authors add the VOC plot. Besides my major point above, I also have several minor comments. 1) On page 1 in abstract (line 18), "wall loss of primary gaseous pollutants was negligible, . . .", please add "was negligible compared with the replenish rate by gas exchange". Because other readers who don't have gas exchange might also think it is negligible in their chamber, which will cause confusion. 2) On page 3 (line 100), is "polytetrafluoroethylene" PTFE? PFA should be "Perfluoroalkoxy alkane". 3) On page 6 (line 207), "The transmission . . . were measured using a Fourier Transform Infrared Spectroscopy (FTIR) system". However, in the following context, the authors are discussing about the UV-Vis spectrum of the Teflon film and acrylic shell. Did I miss something? Or does the authors actually mean "UV-Vis" system? 4) On page 12 (line 444), Fig. 11a and Fig. 11b should be Figure 10a and Figure 10b. Please correct that. 5) On page 13 (line 473, and equation 3), if some of SOA components are not

very soluble in water, the updated formula in reference (Petters and Kreidenweis 2008) (formula 10) should be used. Otherwise, based on the $\kappa$ of 0.04, the average molecular weight of the coating material is about 450 g mol-1. 6) This is not a comment but a discussion. In the end of "Section 4", the authors discussed about the relationship between $\kappa$ and O/C ratio. In the reference (Jimenez et al. 2009), Figure 3 shows at around O/C = 0.5, the $\kappa$ is about 0.12 for ambient (including Mexico city, Jungfraujoch, Hyytilala SV-OOA, Hyytiala LV-OOA) and $\alpha$-pinene SOA, but about 0.16 for Trimethylbenzene (TMB) SOA. All the cases have larger $\kappa$ values than reported here (0.04). However in another reference (Massoli et al. 2010), Figure 2 shows the $\alpha$-pinene and m-xylene SOAs have $\kappa$ value about 0.14 at O/C=0.5 , but TMB SOA has $\kappa$ value of only 0.04, consistent with the value (0.04) reported here. If we believe the latter reference is correct, does that mean the VOC source in Beijing is more TMB like? O/C could not be used as the only parameter for predicting $\kappa$ as discussed. This could also suggest the VOC source in Beijing is different from other places, such as Mexico City.

Overall, I think it is a good manuscript. I support the publication of the manuscript after my major comment and five minor comments are addressed. Reference

Jimenez, J. L., Canagaratna, M. R., Donahue, N. M., Prevot, A. S. H., Zhang, Q., Kroll, J. H., DeCarlo, P. F., Allan, J. D., Coe, H., Ng, N. L., Aiken, A. C., Docherty, K. S., Ulbrich, I. M., Grieshop, A. P., Robinson, A. L., Duplissy, J., Smith, J. D., Wilson, K. R., Lanz, V. A., Hueglin, C., Sun, Y. L., Tian, J., Laaksonen, A., Raatikainen, T., Rautiainen, J., Vaattovaara, P., Ehn, M., Kulmala, M., Tomlinson, J. M., Collins, D. R., Cubison, M. J., E., Dunlea, J., Huffman, J. A., Onasch, T. B., Alfarra, M. R., Williams, P. I., Bower, K., Kondo, Y., Schneider, J., Drewnick, F., Borrmann, S., Weimer, S., Demerjian, K., Salcedo, D., Cottrell, L., Griffin, R., Takami, A., Miyoshi, T., Hatakeyama, S., Shimono, A., Sun, J. Y., Zhang, Y. M., Dzepina, K., Kimmel, J. R., Sueper, D., Jayne, J. T., Herndon, S. C., Trimborn, A. M., Williams, L. R., Wood, E. C., Middlebrook, A. M., Kolb, C. E., Baltensperger, U., Worsnop, D. R. (2009). Evolution of Organic Aerosols in the Atmosphere. Science 326:1525-1529.

Massoli, P., Lambe, A. T., Ahern, A. T., Williams, L. R., Ehn, M., Mikkilä, J., Canagaratna, M. R., Brune, W. H., Onasch, T. B., Jayne, J. T., Petäjä, T., Kulmala, M., Laaksonen, A., Kolb, C. E., Davidovits, P., Worsnop, D. R. (2010). Relationship between aerosol oxidation level and hygroscopic properties of laboratory generated secondary organic aerosol (SOA) particles. Geophys. Res. Lett. 37:L24801.

Petters, M. D. and Kreidenweis, S. M. (2008). A single parameter representation of hygroscopic growth and cloud condensation nucleus activity – Part 2: Including solubility. Atmos. Chem. Phys. 8:6273-6279.

Please also note the supplement to this comment:
http://www.atmos-chem-phys-discuss.net/acp-2017-370/acp-2017-370-RC1-supplement.pdf

---

## Referee Comment (RC2) · Anonymous Referee #2 · 12 Jun 2017

The manuscript investigated the changes in the hygroscopicity of elemental carbon (EC) aerosol during aging in QUALITY chamber where the gaseous species in ambient air constantly diffused into the reaction chamber and products from their photochemical reactions interacted with mono-dispersed EC seed particles. The variations in EC particle size, mass, chemical composition, hygroscopicity and CCN properties were monitored throughout the aging processes. This study provides a unique perspective among chamber studies because the EC seed aerosol was exposed to photochemical oxidation products from an environment that closely mimics the actual ambient air.

Evaluating the hygroscopicity of EC has been a challenge for the atmospheric sciences research community due to the complex morphology changes. Following the method of using mono-dispersed seed aerosol (Qiu et al. ES&T, 2012 and Khalizov et al. ES&T, 2013), this study provided new insights into this complex research problem. The current manuscript also nicely supplements a recent publication from the same authors (Peng et al. PNAS, 2016) by providing detailed evaluation of the the QUALITY chamber. The characterization of the QUALITY chamber experiments appeared to be thorough and the results appeared to be reliable. The topic is relevant to the scope of the journal of Atmospheric Chemistry and Physics and should be considered for publication.

A few comments are provided to facilitate the further improvement of the manuscript: (1) Recent studies showed (for example, Chen et al. Geophys. Res. Lett. 2016, 43, 11080) that the morphology of nascent EC particles can be highly sensitive to even a small change in the organic coating on the primary spheres. As a result, it is prudent to provide experimental data to show that all the EC seed particles started with similar morphology and chemical composition. What was the thickness (or weight percent) of volatile organics in the seed particles at the beginning of each chamber aging experiment? (2) Evaluating the hygroscopic changes of EC has been a challenge. It has been proposed (for example, Qiu et al. ES&T, 2012) that the hygroscopicity of the coating materials can be measured or estimated to reflect the real hygroscopic growth of the EC particles. The authors may consider incorporating such approach to covert the apparent HGF into the real HGF. Furthermore, I wonder if similar approach can be developed for the CCN data to estimate the real $\kappa$ values of aged EC particles. (3) What was the final size of the new particles from nucleation in the chamber? It could be a concern if the nucleation in the chamber was fast and the new particles grew too fast into the size range of the aged soot particles and hence interfere with the property measurements.

---

## Author Comment (AC1) · 17 Jul 2017

We thank the referee for his/her careful and critical review of our paper. The following are our responses to the referee's comments.

In the manuscript the authors measured several physicochemical properties, including size, density, chemical composition, hygroscopicity, and Cloud Condensation Nuclei (CCN) activity of coated Black Carbon (BC) particles as a function of aging (coating with condensable species from photo-oxidation of VOCs) in a quasi-atmospheric

aerosol evolution study (QUALITY) chamber using a suite of instruments, such as differential mobility analyzer, aerosol particle mass analyzer, High Resolution – Time of Flight – Aerosol Mass Spectrometer (HR-ToF-AMS), Humidified Tandem Differential Mobility Analyzer (HTDMA), and a CCN Counter (CCNC). Their results show that under ambient condition in Beijing, the BC undergoes rapid growth to $77\pm33$ nm coating thickness with an average growth rate of $26\pm11$ nm h-1. The O/C ratio of the SOA coating is 0.5, lower than ambient level, indicating the lack of aqueous phase oxidation inside QUALITY chamber. The hygroscopic parameter is about 0.035 - 0.040 as measured by HTDMA and CCNC, suggesting that the initial photochemical aging of BC particles does not appreciably alter the particle hygroscopicity in Beijing. This study and the data provided are quite extensive. The most valuable addition of this manuscript is the "close to ambient (Beijing)" aging condition in the QUALITY chamber. The author showed in Figure 3 that the O3, NOx, CO, and SO2 concentrations inside and outside of the chamber is close. However, the VOCs plot inside and outside of the chamber is missing. The authors also mentioned they applied heater, drier, alumina spherules coated with potassium permanganate (KMnO4), and activated charcoal to remove the VOCs, H2S, SO2, NOx, and O3. It would be nice if the authors could also show the VOCs are efficiently removed down to sub-ppb level while injecting the BC particles into the chamber. Did the authors use proton-transfer-reaction mass spectrometer (PTR-MS) to monitor the VOCs inside and outside the chamber? Since the coating is caused by condensable species from photo-oxidation of VOCs and "close to ambient" condition is an important part of the manuscript, I would suggest the authors add the VOC plot.

We thank the reviewer very much for the insight suggestion. We have added another figure (as Figure 3 in current version), which demonstrates the comparison of VOCs concentration between chamber and ambient air before the BC aging experiment. Corresponding discussion is also added in the text as (Line 231): "To investigate the VOCs concentration inside the reaction chamber after the injection of BC particles, both the chamber and ambient air were sampled with VOC canisters just before the BC aging experiment started. These canisters were then analyzed by a gas chromatographymass spectrometer / flame ionization detector (GC-MS/FID, HP inc.) system (Liu et al., 2008). The concentrations of VOCs containing 4 or more carbons are illustrated in Figure 3. Slightly higher concentrations of several VOCs in the QUALITY chamber, e.g., n-butane, n-pentane, toluene, were observed compared with those in the ambient air, duo to the co-injection of a small amount of VOCs together with BC particles into the chamber. Nevertheless, the average increase of the VOC concentrations was only 16% or 0.1 ppb for all focused VOCs species, with the largest increase of 35% or 0.36 ppb, suggesting the insignificant influence of soot burner on VOCs concentrations and SOA formation in the chamber. "

Besides my major point above, I also have several minor comments. 1) On page 1 in abstract (line 18), "wall loss of primary gaseous pollutants was negligible, : : :", please add "was negligible compared with the replenish rate by gas exchange". Because other readers who don't have gas exchange might also think it is negligible in their chamber, which will cause confusion.

We thank the reviewer for this suggestion. The sentence has been modified as suggested.

2) On page 3 (line 100), is "polytetrafluoroethylene" PTFE? PFA should be "Perfluoroalkoxy alkane".

We are sorry for such mistake. The "polytetrafluoroethylene" should be "Perfluoroalkoxy alkane". It has been revised in the manuscript.

3) On page 6 (line 207), "The transmission : : : were measured using a Fourier Transform Infrared Spectroscopy (FTIR) system". However, in the following context, the authors are discussing about the UV-Vis spectrum of the Teflon film and acrylic shell. Did I miss something? Or does the authors actually mean "UV-Vis" system?

We thank the author for pointing out his mistake. The sentence has been revised as "The transmission efficiencies of each material were measured using an ultraviolet-

visible (UV-Vis) spectrophotometry (PerkinElmer Inc., model 552)."

4) On page 12 (line 444), Fig. 11a and Fig. 11b should be Figure 10a and Figure 10b. Please correct that.

It has been revised. Thanks.

5) On page 13 (line 473, and equation 3), if some of SOA components are not very soluble in water, the updated formula in reference (Petters and Kreidenweis 2008) (formula 10) should be used. Otherwise, based on the kappa of 0.04, the average molecular weight of the coating material is about 450 g mol-1.

We agree with the reviewer that the updated formula in reference (Petters and Kreidenweis 2008) (formula 10) is more accurate to calculation the kappa value when coating material is not highly hygroscopic. However, the calculation requires the solubility of SOA, which is impossible to be obtained duo to the variable VOC precursors and complex photochemical reactions in the ambient condition. Nevertheless, we added discussion on the uncertainty to use of this mixing rule as "Since the SOA formed inside the chamber was not highly hygroscopic, some of the SOA components might not be able to solve in water droplets (Petters and Kreidenweis, 2008), lead to the underestimation of the $\kappa$ values of the coating materials in this study."

6) This is not a comment but a discussion. In the end of "Section 4", the authors discussed about the relationship between _ and O/C ratio. In the reference (Jimenez et al. 2009), Figure 3 shows at around O/C = 0.5, the _ is about 0.12 for ambient (including Mexico City, Jungfraujoch, Hyytilala SV-OOA, Hyytiala LV-OOA) and _-pinene SOA, but about 0.16 for Trimethylbenzene (TMB) SOA. All the cases have larger _ values than reported here (0.04). However, in another reference (Massoli et al. 2010), Figure 2 shows the _-pinene and m-xylene SOAs have _ value about 0.14 at O/C=0.5, but TMB SOA has _ value of only 0.04, consistent with the value (0.04) reported here. If we believe the latter reference is correct, does that mean the VOC source in Beijing is more TMB like? O/C could not be used as the only parameter for predicting _ as

discussed. This could also suggest the VOC source in Beijing is different from other places, such as Mexico City.

The reviewer made a very interesting point to examine the application of this study. However, we don't have enough evidence to draw such conclusion yet. First, "The O/C ratio of the coating SOA is 0.5 in our experiment, corresponding to the O/C ratio of approximately 0.4 in Jimenez et al. (2009) and Massoli et al. (2010) duo the utilize of updated AMS calibration method in this study." The $\kappa$ value at O/C=0.4 will be lower than that provided above by the reviewer. Second, the $\kappa$_HGF of $\alpha$-pinene and $\alpha$-pinene/xylene mixture is consistent with our $\kappa$_HGF at same O/C ratio. Third, in the measurements at other sites around the world, e.g., Mexico City, Jungfraujoch, Hyytilala, the measured $\kappa$ value of organics were for the total OM in the atmosphere, including both primary and secondary OM with different aging degree (from hours to days) through both photochemical and heterogenous pathways. In this study, however, the SOA is mainly from photochemical reactions with aging time scale of only a few hours. Therefore, we can't conclude that the SOA hygroscopiciy in Beijing is different from other cities at the moment. But we believe that this is an interesting aspect and we would like to focus this in our future studies.  

---

## Author Comment (AC2) · 17 Jul 2017

We thank the referee for his/her careful and critical review of our paper. The following are our responses to the referee's comments.

The manuscript investigated the changes in the hygroscopicity of elemental carbon (EC) aerosol during aging in QUALITY chamber where the gaseous species in ambient air constantly diffused into the reaction chamber and products from their photochemical reactions interacted with mono-dispersed EC seed particles. The variations in EC

particle size, mass, chemical composition, hygroscopicity and CCN properties were monitored throughout the aging processes. This study provides a unique perspective among chamber studies because the EC seed aerosol was exposed to photochemical oxidation products from an environment that closely mimics the actual ambient air. Evaluating the hygroscopicity of EC has been a challenge for the atmospheric sciences research community due to the complex morphology changes. Following the method of using mono-dispersed seed aerosol (Qiu et al. ES&T, 2012 and Khalizov et al. ES&T, 2013), this study provided new insights into this complex research problem. The current manuscript also nicely supplements a recent publication from the same authors (Peng et al. PNAS, 2016) by providing detailed evaluation of the the QUALITY chamber. The characterization of the QUALITY chamber experiments appeared to be thorough and the results appeared to be reliable. The topic is relevant to the scope of the journal of Atmospheric Chemistry and Physics and should be considered for publication. A few comments are provided to facilitate the further improvement of the manuscript: (1) Recent studies showed (for example, Chen et al. Geophys. Res. Lett. 2016, 43, 11080) that the morphology of nascent EC particles can be highly sensitive to even a small change in the organic coating on the primary spheres. As a result, it is prudent to provide experimental data to show that all the EC seed particles started with similar morphology and chemical composition. What was the thickness (or weight percent) of volatile organics in the seed particles at the beginning of each chamber aging experiment?

We thank the reviewer for the suggestion. In our study, the fresh BC particles were highly fractal. For three initial mobility diameters (Dm) of about 100 nm, 150 nm, and 200 nm, the ranges of effective density for fresh BC particles in each experiment are 0.43-0.50 g cm−3, 0.34-0.34 g cm−3, and 0.24-0.32 g cm−3, respectively (Table 2 of the manuscript). Accordingly, the dynamic shape factors (DSF) of 100 nm, 150 nm, and 200 nm BC particle are 2.11-2.30, 2.54-2.55 and 2.44-2.85, respectively. This suggests the highly fractal morphology of the fresh BC particles, and the consistency of fresh BC properties in different experiment. Also, as the BC particles formed from

the burner were heated to 300  C before introduced into the chamber, most of the organic coatings were removed. The combined measurement of particle size distribution, density and chemical composition (AMS) shows that organics accounted for less than 10% of fresh BC mass concentration. The single scattering albedo (SSA) of fresh BC particles was only 0.1, further confirming the purity of BC particles. We modified two places in the manuscript to make this clear: Line 132, "The combined measurement of particle size distribution, density and chemical composition exhibited that organics accounted for less than 10% of fresh BC mass concentration. Moreover, the single scattering albedo (SSA) of fresh BC particles was only 0.1, further confirming that few organic coatings exited on fresh BC particles." Line 238, "For three initial $D_m$ of 100 nm, 150 nm, and 200 nm, the ranges of effective density of fresh BC particles in each experiment were 0.43-0.50 g cm$-3$, 0.34-0.34 g cm$-3$, and 0.24-0.32 g cm$-3$, respectively, indicating highly fractal BC aggregates (Zhang et al., 2008). The small variation for particles with the same $D_m$ also demonstrates the consistency of fresh BC morphology in different experiments."

(2) Evaluating the hygroscopic changes of EC has been a challenge. It has been proposed (for example, Qiu et al. ES&T, 2012) that the hygroscopicity of the coating materials can be measured or estimated to reflect the real hygroscopic growth of the EC particles. The authors may consider incorporating such approach to covert the apparent HGF into the real HGF. Furthermore, I wonder if similar approach can be developed for the CCN data to estimate the real _ values of aged EC particles.

The reviewer made a very good point here. As discussed in the manuscript, the apparent HGF may underestimate the hygroscopicity of BC particles when particles are not spherical shape. We attribute this the major reason of the difference between $\kappa\_CCN$ and $\kappa\_HTDMA$ in the beginning of experiment. However, duo to the fast growth of BC particles in our study, the morphology of BC particles shifted from fractal to spherical shape within a very short time. Therefore, the apparent HGF at the end of experiments can represent real HGF of particles. Besides, in the CCN data processing, the particle

mass equivalent diameter was used as the particle dry diameter in the calculation of $\kappa\_CCN$ value by Equ (10). Therefore, the $\kappa\_CCN$ should reflect the real hygroscopicity of BC particles from this point of view.

(3) What was the final size of the new particles from nucleation in the chamber? It could be a concern if the nucleation in the chamber was fast and the new particles grew too fast into the size range of the aged soot particles and hence interfere with the property measurements.

Nucleation occurred in the chamber during several experiments of this study. Moreover, the newly formed particles grew along with BC particles inside the chamber. However, as shown in the Figure below, the difference in the peak diameter between BC particles and nucleation particles was always larger than 80 nm for all experiments. Therefore, the influence of nucleation particles to the measurement of BC properties was insignificant.

Figure 1. Peak diameter of BC and nucleation particles inside the chamber during experiments with initial BC diameter of 100 nm (A), 150 nm (B), and 200 nm, respectively.

[Figure]